# On reactive Ion Etching of Parylene-C with Simple Photoresist Mask for Fabrication of High Porosity Membranes to Capture Circulating and Exfoliated Tumor Cells

**DOI:** 10.3390/mi15040521

**Published:** 2024-04-13

**Authors:** Inad Rabadi, David Carpentieri, Jue Wang, Frederic Zenhausern, Jian Gu

**Affiliations:** 1Center for Applied NanoBioscience and Medicine, The University of Arizona College of Medicine, Phoenix, AZ 85004, USA; irabadi@arizona.edu (I.R.); fzenhaus@arizona.edu (F.Z.); 2Department of Basic Medical Sciences, The University of Arizona College of Medicine, Phoenix, AZ 85004, USA; 3Phoenix Children’s Hospital, Phoenix, AZ 85016, USA; dcarpentieri@phoenixchildrens.com; 4Dignity Health-Cancer Institute at St. Joseph’s Hospital and Medical Center, Phoenix, AZ 85004, USA; jue.wang@utsouthwestern.edu; 5Honor Health Research Institute, Scottsdale, AZ 85258, USA

**Keywords:** high porosity parylene membrane, circulating/exfoliated tumor cell capture, reactive ion etching

## Abstract

A high porosity micropore arrayed parylene membrane is a promising device that is used to capture circulating and exfoliated tumor cells (CTCs and ETCs) for liquid biopsy applications. However, its fabrication still requires either expensive equipment or an expensive process. Here, we report on the fabrication of high porosity (>40%) micropore arrayed parylene membranes through a simple reactive ion etching (RIE) that uses photoresist as the etching mask. Vertical sidewalls were observed in etched parylene pores despite the sloped photoresist mask sidewalls, which was found to be due to the simultaneous high DC-bias RIE induced photoresist melting and substrate pedestal formation. A theoretical model has been derived to illustrate the dependence of the maximum membrane thickness on the final pore-to-pore spacing, and it is consistent with the experimental data. A simple, yet accurate, low number (<50) cell counting method was demonstrated through counting cells directly inside a pipette tip under phase-contrast microscope. Membranes as thin as 3 μm showed utility for low number tumor cell capture, with an efficiency of 87–92%.

## 1. Introduction

The relevance of using clinical liquid biopsies for cancer detection and prognosis is growing and gaining FDA regulatory approval, but there is still a need for more efficient isolation methods to be developed in order to validate their usefulness in clinical practice. Circulating and exfoliated tumor cells (CTCs and ETCs) are promising liquid biopsy targets for cancer diagnosis, monitoring, and management [1,2]. However, it is challenging to capture these rare cells from large volumes of blood or other relevant bodily fluids. Different methods have been explored, and they can be divided into two main categories. One category is antibody-based immunecapture through cancer cell surface markers, such as epithelial cell adhesion molecule (EpCAM). These methods include the first Food and Drug Administration (FDA) approved Cell Search assay through magnetic-activated cell sorting (MACS) [3] and the microfluidic herringbone graphene oxide chip [4]. The limitation, however, is the possibility of missing cancer cells that are of mesenchymal origin or under epithelial mesenchymal transition (EMT).

The second category of methods does not depend on specific cell surface marker, instead, they separate cancer cells by their physical properties, such as size, deformability, and electrical properties. Different mechanisms and their combinations in microfluidic format have been reported, such as size-based filtration [5,6], titled-angle standing surface acoustic waves (taSSAW) [7], deterministic lateral displacement (DLD) [8,9], dielectrophoresis [9,10], inertial Dean Flow [11,12], etc. Among these methods, membrane-based filtration has been shown to be a simple way of capturing CTCs/ETCs by size, with high efficiency and high throughput (~ ml/min), from large volumes of samples [13,14]. Compared to track-etched membranes, microfabricated membranes with arrayed pores have the advantage of achieving a precisely controlled pore size, geometry, and density with high porosity that enables high throughput and high efficiency CTC/ETC capture [14,15,16].

Parylene is a promising membrane material for CTC/ETC capture because of its biocompatibility, chemical resistance, mechanical strength, and optical clarity [16]. There are multiple ways to pattern parylene, such as wet etching [17], heat or electrical arc [18], laser ablation [19,20], hot embossing or micromolding [21,22], and plasma etching [23]. Among them, wet etching, heat, and electrical arc do not have good dimension control and are not suitable for the micropatterning of porous membranes for CTC/ETC capture. Laser ablation has demonstrated patterning resolution ~3 μm [19], but it is a serial process with low throughput. Hot embossing or micromolding are high throughput processes, and micromolding has been used to fabricate high porosity parylene membranes for CTC/ETC capture [15,24]. However, parylene mold separation can be an issue, and the Si mold must be sacrificed to release the porous membrane [24], making it an expensive process. Finally, dry plasma etching has been used to pattern parylene films with good dimension control. However, metal etching mask [16,25] or deep etching with switched chemistry [23] are usually used, which requires expensive metal deposition or deep plasma etching equipment. On the other hand, simple reactive ion etching (RIE) with photoresist as etching mask was also used to pattern parylene [23,26], but has not been used to make porous filter membranes for CTC/ETC capture.

In this work, we demonstrate our efforts of patterning high porosity (>40%) micropore arrayed parylene membranes through simple O_2_ plasma RIE with a photoresist etching mask (Figure 1). Vertical sidewalls of parylene pores were observed even with sloped photoresist mask sidewalls, which was attributed to simultaneous high DC-bias RIE induced photoresist melting and substrate pedestal formation. A theoretical model was derived between the maximum membrane thickness and the final pore-to-pore spacing, which is in agreement with experimental data. Finally, with a simple and accurate low cell counting method, high capture efficiency of low number spiked tumor cells was demonstrated using the fabricated porous parylene membranes with a thickness of only 3 μm. These devices would provide a high throughput and cost-effective isolation method for broader use in clinical research applications.

## 2. Materials and Methods

### 2.1. Pattern Design and Photomask Fabrication

Close-packed hexagonal-shaped micropore arrays were designed with pore size *D* of 8, 10, 12, or 40 µm, and pore-to-pore spacing *S* of 4 or 6 µm using AutoCAD (Autodesk, Inc., San Rafael, CA, USA) (Figure 2). The porosity of the close-packed hexagonal-shaped pore array can be derived as [*D*/(*D*+*S*)]^2^ and calculated to be 32.7–75.6% for our array designs (See Appendix A). The porosity is expected to be further increased due to lateral etching of the pore-to-pore spacing during fabrication (arrays with *S*/*D* ratio < 58.1% will have a porosity over 40%). For each array, the array area was 13 mm × 13 mm. The design was sent out to Photo Sciences (Torrance, CA, USA) for the fabrication of Chrome-on-Soda Lime photolithography mask.

### 2.2. Fabrication of Microarrayed Porous Parylene Membrane

Parylene-C dimer (DPX-C) was purchased from Specialty Coating Systems, Indianapolis, IN, USA. Parylene-C thin films with different thicknesses were deposited on 4” Si wafers using a PDS 2010 Parylene-Coater (Specialty Coating Systems, Indianapolis, IN, USA). The film thickness was controlled by the amount of dimer used.

AZ10XT-520cP positive photoresist was purchased from Integrated Micro Materials, Argyle, TX, USA, and it was used as the RIE etching mask. Photoresist spin coating was done by a Laurell WS-650 spin coater (Laurell Technologies Corp., North Wales, PA, USA). A spin coating calibration curve was established to obtain desired photoresist thickness. After a soft bake, the photoresist was exposed to ultraviolet (UV) light through the chromium photomask using an HTG mask aligner (San Jose, CA, USA) by contact photolithography. The photoresist was then developed in MIF-300 developer for various times at room temperature, and then rinsed in deionized (DI) water, and then dried by N_2_ gas.

The RIE etching step was performed using a Plasmalab 80 Plus RIE etcher (Oxford Instruments, Bristol, UK). Oxygen plasma with different power, flow rate, and gas pressure was used.

The etching performance characterization including film thicknesses, vertical and lateral etching rate, size, and shape, and the sidewall profile of the etched micropore arrays was measured using a Dektak 3030 profilometer (Veeco (Sloan/Dektak),Bruker, Billerica, MA, USA), an Olympus BH2-UMA bright field optical microscope (Olympus, Center Valley, PA, USA), and a high-resolution Hitachi S4700 field emission scanning electron microscope (FESEM, Hitachi America, Ltd., Santa Clara, CA, USA; located at the NanoFab core facility at Arizona State University).

It is challenging to image the cross section of etched Parylene-C pores because Parylene-C is ductile under cleavage at room temperature and does not give a clean cross-sectional surface. To address this issue, the fabricated membrane on Si wafer was submerged into liquid nitrogen for 3 to 5 s before cleaving the etched membrane to give a better cross-sectional surface. For FESEM imaging, the samples were coated with a layer of 10–20 nm Au/Pd for top and cross-sectional views using a Cressington 108 auto sputter coater (Cressington Scientific Instruments, Walford, UK) to eliminate charge buildup during FESEM observations.

### 2.3. Spiked Tumor Cell Capture Using Fabricated Parylene-C Microporous Membrane

Osteosarcoma cell-line SJSA-1 cells (ATCC, Cat# CRL-2098) were cultured in a T75 cell-culture flask with RPMI 1640 medium containing 10% Fetal bovine serum (FBS, Gibco, ThermoFisher, Waltham, MA, USA), and 1% penicillin-streptomycin (Gibco, Thermo Fisher, USA). Cells were seeded at a subculture ratio of 1:10 and regularly passed about once a week (approximately 80% confluent). Culture medium was exchanged every third day and cells were used within 20 passes.

For characterizing low number (<50) cell counting inside a 10 μL pipette tip (Neptune, Cat# BT10), Countess Cell Counting Chamber Slides (ThermoFisher, Cat# C10228) and Nikon TS-100 phase-contrast microscopewere used. For high cell number calculations, cell density was measured by a Countess (ThermoFisher, MA, USA) automated cell counter.

For immunofluorescent imaging of captured tumor cells, CellTracker-Red (ThermoFisher, Cat# C34552) was used to stain the cytoplasm of SJSA-1 cells before the capture, according to the vendor’s instruction. After the capture, the cells were fixed on the Parylene-C membrane inside the capture device by 4% paraformaldehyde with 10 mL PBS wash, then stained by 300 nM DAPI (life Technologies Corporation, Eugene, OR, USA) solution with 20 mL PBS wash. The device was then disassembled, and the membrane was transferred to a glass slide to be mounted under a cover glass using CoverGrip Sealant (BIOTIUM, Fremont, CA, USA, Cat#23005). Finally, the sample was imaged using a Zeiss AXIO Imager M2 Epifluorescent Microscope (Carl Zeiss Microscopy, LLC, White Plains, NY, USA) in the Biomedical Imaging Core at the University of Arizona College of Medicine-Phoenix. The collected images were then analyzed using Zen 2.6 lite software to count the captured cells.

## 3. Results

### 3.1. RIE Fabrication of Micropore Arrayed Parylene Membrane

Figure 3a shows the detailed process flow of fabricating parylene microporous membranes using RIE with a photoresist etching mask, coined as the Simple Parylene Etching (SPE) process. First, a parylene thin film is deposited on a Si wafer. Then, the photoresist etching mask was patterned through spin coating onto the parylene film, followed by photolithography exposure by a contact mask aligner and resist development. With the patterned photoresist mask, the parylene film was etched by O_2_ plasma RIE. Any remaining photoresist mask was removed by acetone before the parylene microporous membrane was peeled away from the Si wafer.

With the SPE method, parylene membranes with microporous arrays were successfully fabricated. Figure 3b shows a 4″ Si wafer with microporous arrays etched into a parylene membrane with photoresist as the mask, and a parylene membrane with a microporous array peeled off from the Si substrate.

To understand the etching process, the sidewall profiles of the pores in photoresist and parylene before and after etching were imaged by FESEM, as shown in Figure 3c. It shows that the pores in the photoresist etching mask have a positive sloped sidewall. This is common for contact photolithography due to the scattering of the light at the pattern edge [27]. However, a surprising observation is that a vertical sidewall profile in the final etched parylene pore was formed, which contrasts with previous literature report that the sloped sidewall in the photoresist mask would be transferred into the parylene layer [28].

### 3.2. Condition for Vertical Sidewall Formation in Etched Parylene Pores

A vertical sidewall is generally preferred during pattern transfer for pattern dimension control. To investigate the cause of the vertical sidewall formation, a test grating mask with different line/space patterns was used to pattern a 10-μm-thick photoresist mask for a 4.5-μm-thick parylene layer. Then time lapse imaging of the cross-section of the etched line/space pattern was performed by taking one piece of sample out every 5 min of etching to cleave for cross-sectional imaging. The top row of Table 1 lists the Vertical Sidewall recipe that was used. Figure 4a shows the cross sections of a 20-μm-period line/space pattern at different time points etched using the recipe. It shows that the photoresist line had a typical sloped sidewall (~70–75 degrees) after photolithography. However, after 5 min of etching, the photoresist shape changed from a trapezoid to a circular cap shape, and the quasi-vertical sidewall of the circular cap resulted in a vertical sidewall of the etched parylene.

The transition of the photoresist shape is consistent with the surface tension caused shape change after photoresist melting [29,30]. It is also well known that the ion bombardment of the RIE process can heat the sample surface [31,32]. Therefore, we hypothesize that ion bombardment from the Vertical Sidewall RIE recipe have high enough energy to melt the photoresist mask that causes vertical sidewalls in parylene pores. To validate this hypothesis, another RIE recipe with lower DC bias for lower ion bombardment energy (Table 1 bottom row) was used to etch the same line/space pattern (Figure 4b). Indeed, no photoresist melting were observed, and the sloped sidewall in photoresist was transferred to the parylene film.

To further understand if photoresist melting without plasma can also have the same effect for parylene etching, samples with patterned photoresist before RIE were heated on a hotplate at 80–150 °C. However, the photoresist pattern maintained its profile at low temperatures (<110 °C), and at higher temperatures (110 °C or higher) it completely melted and wetted the underlying parylene substrate, resulting in a pattern loss. No circular cap shape of photoresist was formed. This is consistent with previous reports that a pedestal structure is critical in containing the reflow of the photoresist [30].

With all studies mentioned above, we conclude that the high energy ion bombardment RIE caused photoresist melting and pedestal formation at the same time, and resulted in the circular cap shape of photoresist for vertical sidewalls of the etched parylene patterns.

### 3.3. Dependance of Maximum Thickness of Porous Parylene Membrane on Pore-to-Pore Spacing by the Vertical Sidewall SPE Process

It has been reported that having high porosity with narrow pore-to-pore spacing (i.e., *S* smaller than half of the cell diameter) is critical in improving the sample processing throughput, increasing viability of the captured cells, and reducing contaminated cells during CTC/ETC capture [15]. The Vertical Sidewall SPE process offers better pattern dimension control for fabricating parylene micropore arrays with small *S*. However, the sloped photoresist mask due to our photolithography process is expected to put a constraint on the maximum parylene membrane thickness that can be achieved for certain *S*. Thickness of the membrane is important for maintaining the mechanical strength of the membrane. Previously, a fixed membrane thickness of 10 μm has been used [15,16]. It is not clear if a thinner membrane can be used for CTC/ETC capture. In this section, we study the dependency of etched membrane thickness on *S*, and report the possibilities and limitations on fabricating small *S* parylene micropore arrays using the Vertical Sidewall SPE process.

To estimate the dependency, we formulated a mathematical model, as shown in Figure 5a. For the maximum amount of photoresist mask, a triangular photoresist cross section with a slope angle of *α* is assumed after photolithography, and the maximum photoresist height *H_pr,max_* can be expressed as:(1)$$H_{\mathrm{pr},\max}=\frac{S}{2}\times \tan\alpha ,$$

After melting of the photoresist, if we assume the cross-sectional area of the circular cap *A_sp_* is the same as the initial triangular area *A_tr_*, then we have:$$A_{\mathrm{sp}}=\left(\pi -\theta \right)\times {\left(\frac{S/2}{\sin\theta }\right)}^{2}+\frac{S}{2}\times \frac{S/2}{\tan\theta }=\frac{S^{2}}{4}\times \left[\frac{\pi -\theta }{\sin\theta }+\frac{1}{\tan\theta }\right],$$
$$A_{\mathrm{sp}}=\left(\pi -\theta \right)\times {\left(\frac{S/2}{\sin\theta }\right)}^{2}+\frac{S}{2}\times \frac{S/2}{\tan\theta }=\frac{S^{2}}{4}\times \left[\frac{\pi -\theta }{\sin\theta }+\frac{1}{\tan\theta }\right],$$
(2)$$A_{\mathrm{sp}}=A_{tr}\to \left[\frac{\pi -\theta }{\sin\theta }+\frac{1}{\tan\theta }\right]=\tan\alpha ,$$

For a typical 75° slope in our photoresist mask, Equations (1) and (2) leads to a *θ* of 56.74^o^, and the height of the circular cap *H_pr,cap_* can be expressed as:(3)$$H_{\mathrm{pr},\mathrm{cap}}=\frac{S/2}{\sin\theta }+\frac{S/2}{\tan\theta }=0.926*S,$$

With etching selectivity *Se*, parylene vertical and lateral etching rates *R_V,pa_* and *R_L,pa_*, the maximum parylene membrane thickness *H_pa,max_* and final pore-to-pore spacing *S_f_* can be expressed as:(4)$$H_{\mathrm{pa},\max}=\mathrm{Se}\times H_{\mathrm{pr},\mathrm{cap}}\;\mathrm{and}\;S_{f}=S-2R_{L,\mathrm{pa}}*\frac{H_{\mathrm{pa},\max}}{R_{V,\mathrm{pa}}},$$

Solving Equations (3) and (4) with values of *Se*, *R_V,pa_* and *R_L,pa_* from Table 1 gives:(5)$$S_{f}=\frac{H_{\mathrm{pa},\max}}{0.926*\mathrm{Se}}-2R_{L}*\frac{H_{\mathrm{pa},\max}}{R_{V,\mathrm{pa}}}\;\to \;H_{\mathrm{pa},\max}=\frac{1}{\frac{1}{0.926*\mathrm{Se}}-\frac{2R_{L,\mathrm{pa}}}{R_{V,\mathrm{pa}}}}S_{f}=1.39*S_{f}$$
i.e., the maximum parylene membrane thickness is 1.39 times of the final pore-to-pore spacing.

To test our prediction, parylene membranes with differently arrayed pores were fabricated utilizing the Vertical Sidewall SPE process using our mask patterns. The thicknesses of the starting parylene film of ~1, 3, 5, and 10 μm were used. The starting photoresist thickness was twice of the usual parylene film thickness. Different exposure and development times during photolithography were also used to obtain different *S* at the base in the photoresist mask. The membranes with photoresist masks around being completely etched away were used to show the relationship between *H_pa,max_* and *S_f_*, as shown in Figure 5b. A linear fitting through the origin shows that *H_pa,max_* is 1.32 times of the *S_f_*, in good agreement with our theoretical model. Figure 5c–e also show top and cross-sectional/tilted views of three fabricated porous parylene membranes with different *D_f_*/*S_f_* values by FESEM, where *D_f_* is the final size of the pore.

### 3.4. Capture of Spiked Tumor Cells Using SPE Fabricated Porous Membrane

The fabricated parylene micropore array membrane was tested for capturing rare tumor cells spiked in PBS. SJSA-1 osteosarcoma cell line was used, which is expected to be low on epithelial cell adhesion molecule (EpCAM) expression (the main target for affinity-based CTC capture), and it is a good candidate for size-based capture [33,34].

A challenge to characterizing the efficiency of the rare cell capture is the accurate counting of low number of cells (<50). Previously, multiple pieces of literature reported the spiking of low number of cells. However, they used either sophisticated equipment, such as micropipette and precise manipulator [15], or repeated measurement with large variations (up to 20%) [35]; others had vague descriptions of the process [36,37]. Here we report a simple low cell number counting method by diluting the cell to a desired concentration first, then pipetting 10 μL solution and counting the cells directly inside the 10 μL pipette tip under a phase-contrast microscope. Indeed, cells were visible inside the pipette tip (Figure 6a). Focus was adjusted to count all the cells at different focal planes. To verify the accuracy of the counted number, the solution was pipetted into a cell counting chamber slide to count the cells (Figure 6b). Figure 6c shows the results of three counting experiments with cell number ranging from 20 to 32. The number differences between counts inside the pipette tip and the chamber slide were at most 1, indicating a good accuracy to count the cells inside the pipette tip directly.

To test the efficiency of capturing low number SJSA-1 cells, the cells were fluorescently labeled with CellTracker Red, spiked into 10 mL of PBS, and flowed through 3-µm-thick, 9.5-µm-pore, and 4.5-µm-gap parylene membranes (porosity of 46%) at a rate of 3 mL/min. The custom-made membrane holder device and the capture setup are shown in Appendix A. After capturing the cells, the cells were fixed and DAPI stained for the nuclei on the membrane by flowing the reagents through the device. The membrane was then taken out and mounted on a glass slide, and then scanned to count the captured SJSA-1 cells. Figure 7a shows fluorescent image of a SJSA-1 cell captured on the membrane. Figure 7b shows 5 experiments with low spiked cell numbers of 12–31. The captured cell numbers ranged from 11 to 27, and the capture efficiency ranged from 87–92%. This capture efficiency is comparable to others reported in literature [15]. High cell number spiked experiments were also performed. The initial cell numbers were calculated from the measured cell solution density. Figure 7c shows three experiments with high spiked cell numbers of 1000–2000. The capture efficiency ranged from 87.5–115%. We attribute the over 100% efficiency to the statistical distribution of the cells in solution.

## 4. Discussion

Close-packed hexagonal-shaped high porosity parylene membranes were fabricated previously; however, expensive process or equipment was used, such as micromolding that requires scarification of the Si mold [15,24], or the use of a metal deposition facility to utilize metal as the dry etching mask [25]. Our study explored the SPE process to fabricate micropore arrayed parylene membranes using photoresist as the etching mask. Vertical sidewalls of the etched parylene pores were obtained despite an initial sloped photoresist sidewall, which was attributed to the simultaneous plasma melting of the photoresist mask and the substrate pedestal formation during RIE etching at high DC bias. A theoretical model was derived to estimate *H_pa,max_* for certain *S_f_*, and it was in good agreement with the experimental data. An accurate low number cell counting method, performed by counting cells directly inside pipette tips under a phase-contrast microscope, was demonstrated. The spiked low number SJSA-1 cells in PBS were shown to be captured at an efficiency of 87–92% through the use of our fabricated membrane.

Due to the small pore-to-pore spacing of our pattern design, as well as the finite lateral etching rate, the fabricated parylene membranes have high porosity (>40%) in general. The small pore-to-pore spacing, on the other hand, also limits the membrane thickness to less than the traditional 10 μm, and this raised concern on the mechanical strength of the membranes. During our experiments, we did observe membrane distortion when 1-μm-thick membranes were peeled off from the Si wafer substrate. However, this distortion was not observed for membranes ~3 μm or thicker. 3-μm-thick, porous membranes were also successfully used to capture, immunostain, and count spiked tumor cells. Furthermore, the captured cells were also viable, as shown in Appendix A.

Besides its simplicity, SPE is also a versatile process. For example, it is plausible to fabricate high porosity parylene membranes with a small *S_f_* by the low DC-bias SPE process (Table 1) with a photoresist mask having vertical sidewalls. For CTC applications, the white blood cell depletion rate of the fabricated membranes is expected to be improved due to the small pore-to-pore spacing, and this will be tested in future works. It was also reported that conical-shaped holes could improve the purity of CTC capture [38]. The low DC-bias sloped sidewall SPE process would be a simple way to fabricate conical-shaped pore arrays in parylene, and this would be done by taking advantage of the sloped sidewall of the photoresist etching mask. Overall, this technique will improve the toolbox for the rare cell capture for liquid biopsies, which is promising regarding their future prospect for clinical adoption.

## Figures and Tables

**Figure 1 micromachines-15-00521-f001:**
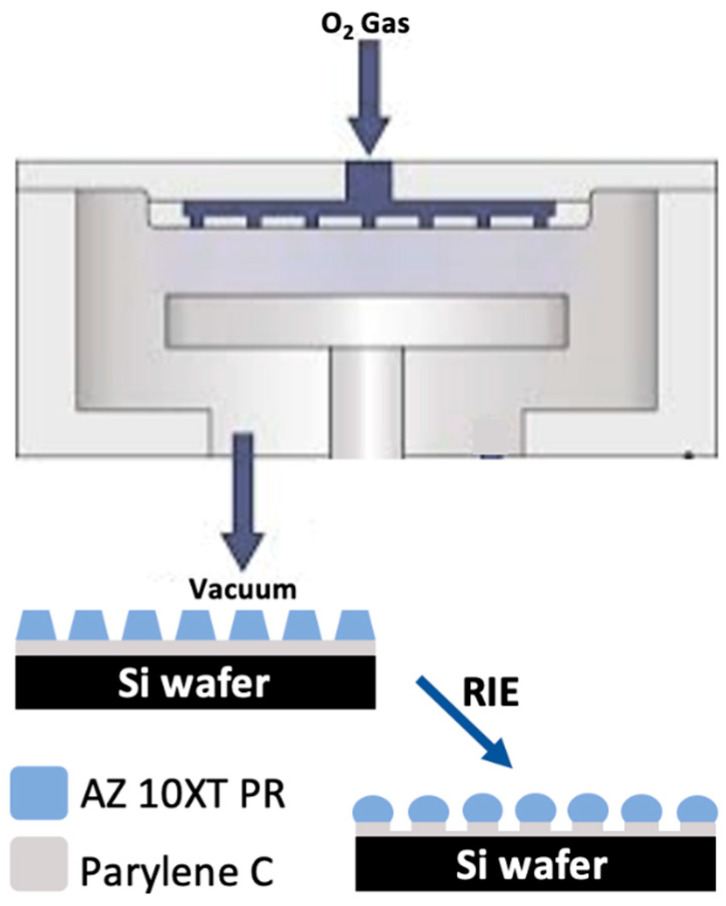
Schematics of a RIE chamber and fabrication of porous parylene membrane by simple RIE using photoresist as a mask.

**Figure 2 micromachines-15-00521-f002:**
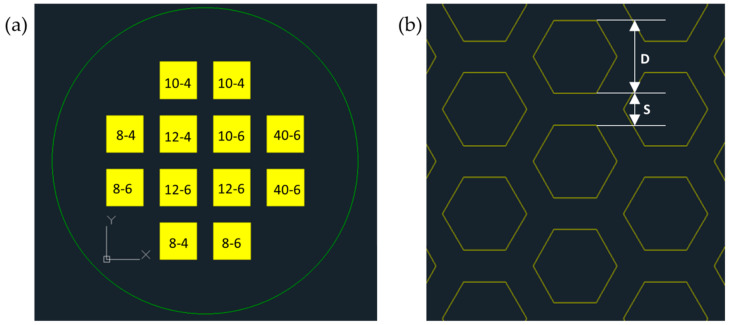
(**a**) Close-packed hexagonal micropore array patterns within a 4″ wafer area. Each array is designated by the pore size *D* and spacing *S* in the format of *D*-*S* in micrometers; (**b**) Zoom-in view of a hexagonal micropore array.

**Figure 3 micromachines-15-00521-f003:**
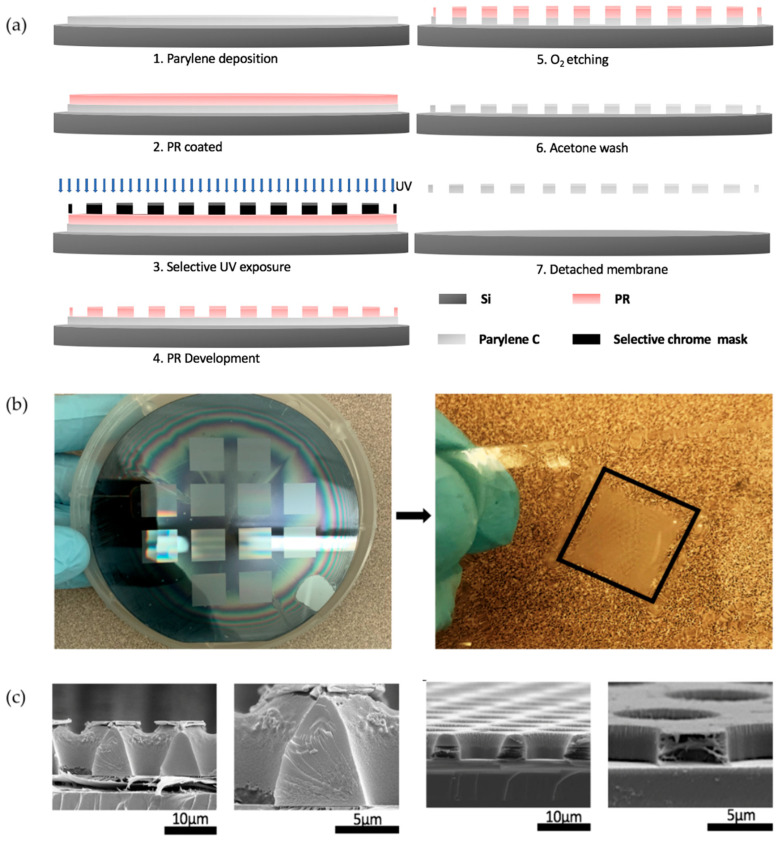
(**a**) Schematic illustration of the parylene porous microarray fabrication by SPE (PR: photoresist); (**b**) parylene membrane on Si wafer after RIE (left), and parylene membrane with a microporous array on a glass slide after successfully peeled off the Si wafer (right); (**c**) photoresist profile after development (two left images) and the final parylene profiles after etching (two right images: with and without photoresist etch mask). Surprising vertical sidewalls were observed in etched parylene pores.

**Figure 4 micromachines-15-00521-f004:**
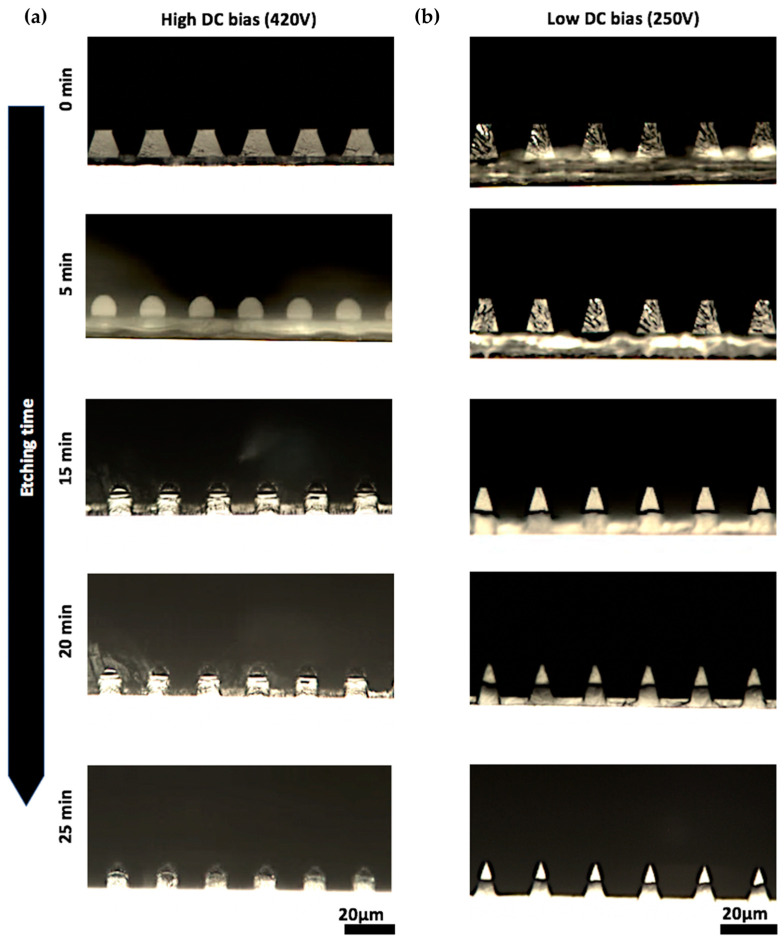
Optical microscope cross-sectional views of (**a**) high (420 V) and (**b**) low (250 V) DC bias RIE recipe etched 20-µm-period grating samples at etching time points of 0, 5, 15, 20, 25 min with 10-µm-thick AZ10XT photoresist mask on 4.5-µm-thick Parylene-C film.

**Figure 5 micromachines-15-00521-f005:**
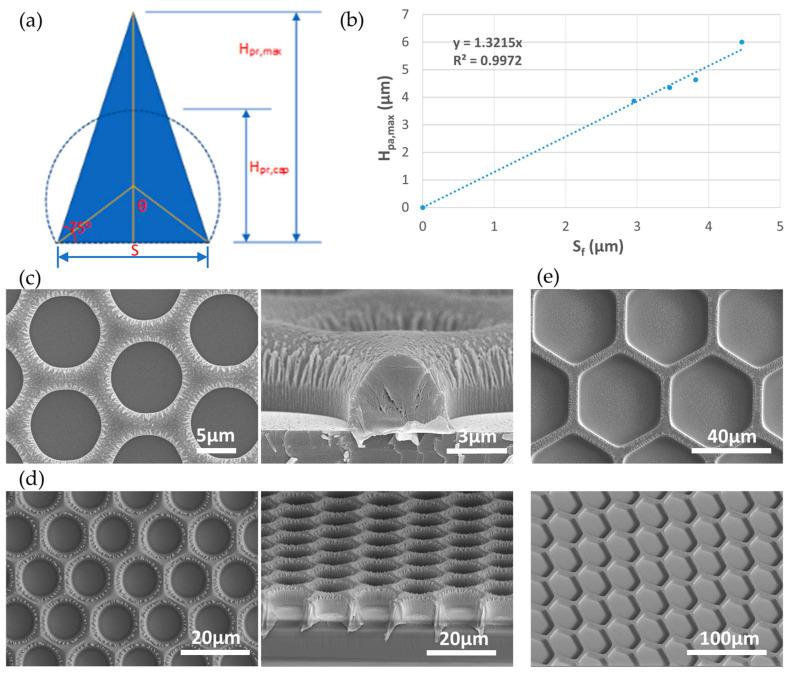
(**a**) Schematics of photoresist shape change due to melting for theoretical analysis of *H_pa,max_* vs. *S_f_*; (**b**) plot of *H_pa,max_* vs. *S_f_* from etched porous parylene membranes; FESEM images of fabricated porous parylene membranes with (**c**) *D_f_*/*S_f_* = 9/3 μm and 3 μm thickness (a thin photoresist mask remains at the top). Left/Right: top/cross-sectional views; (**d**) *D_f_*/*S_f_* = 9.5/4.5 μm and 6 μm thickness (a thin photoresist mask remains at the top). Left/Right: top/cross-sectional views; (**e**) *D_f_*/*S_f_* = 40/6 μm and 6 μm thickness (thin remaining photoresist mask is removed). Top/Bottom: top/tilted views.

**Figure 6 micromachines-15-00521-f006:**
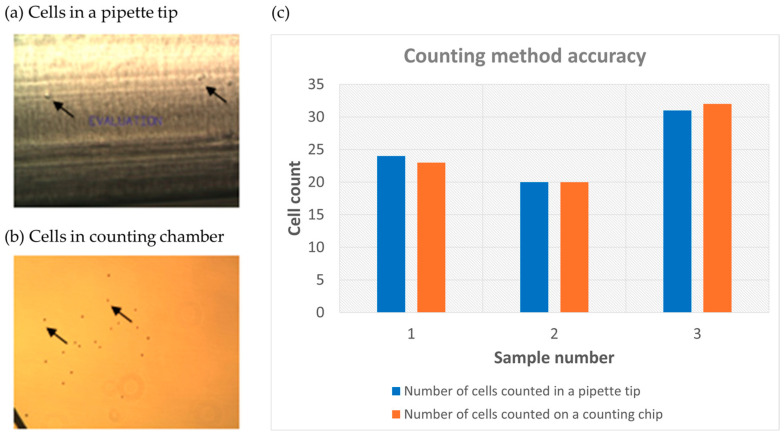
Low number of SJSA-1 cell counting in a 10 µL pipette tip. (**a**) SJSA-1 cells observed by phase-contrast microscopy in the tip; (**b**) same SJSA-1 cells loaded into a counting chamber slide and counted under phase-contrast microscopy; (the arrows show locations of representative cells inside the pipette tip and the counting chamber respectively.) (**c**) three counting results with the numbers from the pipette tip and the numbers from a counting chamber slide.

**Figure 7 micromachines-15-00521-f007:**
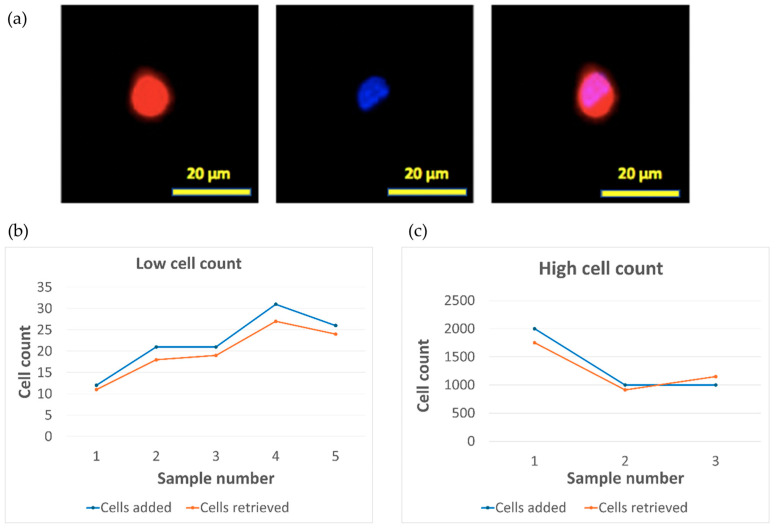
(**a**) CellTracker (**left**), DAPI (**middle**), and combined (**right**) fluorescent images of a capture SJSA-1 cell; Spiked cell capture rates for (**b**) low spiked cell number and (**c**) high spiked cell number.

**Table 1 micromachines-15-00521-t001:** RIE etching recipes for vertical and sloped sidewalls.

Recipe	O_2_ Flow (Sccm)	Pressure (mTorr)	Power (W)	DC Bias (V)	*R_v,pa_* ^1^ (μm/min)	*R_v,pr_* ^1^ (μm/min)	*Se* ^1^ = *R_v,pa_*/*R_v,pr_*	*R_L,pa_* ^1^ (μm/min)
Vertical sidewall	25	200	300	420	0.23	0.25	0.92	0.052
Sloped sidewall	62	500	300	250	0.19	0.205	0.93	0.148

^1^ *R_v,pa_*: parylene vertical etching rate; *R_v,pr_*: photoresist vertical etching rate; *Se*: selectivity; *R_L,pa_*: parylene lateral etching rate.

## Data Availability

Data are contained within the article.

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
