# Peer review of "On reactive Ion Etching of Parylene-C with Simple Photoresist Mask for Fabrication of High Porosity Membranes to Capture Circulating and Exfoliated Tumor Cells"

_micromachines, 2024, doi:10.3390/mi15040521_

Round 1

Reviewer 1 Report

Comments and Suggestions for Authors The manuscript is very interesting and can be published after a minor revision, addressing my following comments:
  • 1. Please add a schematic figure as Figure 1 to describe the point of your work in a quick, easy manner, without jumping to details
  • 2. Given that you have fabricated different geometries, please show different designs as a separate figure

Comments on the Quality of English Language

the Quality of English Language is fine.

Author Response

We appreciate the reviewer’s valuable comments on our manuscript and suggestion of publishing after minor revision.  Here are our replies to the comments:

  1. Please add a schematic figure as Figure 1 to describe the point of work in a quick, easy manner, without jumping to details.

Reply: Thanks for the reviewer’s suggestion. We feel the main point of our work is the simple RIE etching of porous parylene membranes. So we used a modified version of the “Graphic Abstract” to show the schematic of the process as Figure 1 at the end of the “Introduction”. Other figure numbers in the manuscript have also been changed accordingly to be consistent.

  1. Given that you have fabricated different geometries, please show different designs as a separate figure

Reply: Thanks for the reviewer’s comments. We want to clarify that all our pattern designs are close-packed hexagonal-shaped micropore arrays, as shown in Figure 2b (revised manuscript). The different geometries refer to the different Df and Sf values. To give audience a better understanding of the porous parylene membrane fabricated, we also added FESEM images of 3 fabricated porous parylene membranes with different Df/Sf values in Figure 5(c-e).

Reviewer 2 Report

Comments and Suggestions for Authors

This work reports the fabrication of high porosity micropore arrayed parylene membranes by simple reactive ion etching (RIE) using photoresist as the etching mask. A simple cell counting method was demonstrated by counting cells inside a pipette tip under phase-contrast microscope. This work is useful for the community. However, I have several concerns before this manuscript can be accepted. Therefore, in its current form, minor revisions are needed.

1. The authors need to improve the image quality. Some images contain text that is too blurred, such as Fig 5c and 6b-6c.

2. The introduction is not well presented. I recommend the authors to provide a more detailed introduction about different methods for capturing CTC/ETCs or other cells. For example, DLD, dean flow, electric, acoustic, and magnetic methods. 10.1038/nnano.2016.134\ 10.1002/elps.201800459\ 10.1021/acs.analchem.8b05749\ 10.1021/acs.analchem.3c05755\ 10.1039/d2lc01193j\ 10.1021/acs.analchem.1c00312\ 10.1073/pnas.1709210114\ 10.1073/pnas.1504484112\ 10.1039/b903950c

 3.Could this chip be reused again? 

 4. While the manuscript notes improvements over existing techniques, a more concrete comparison with similar designs would provide clearer context and justification for the advancement of this method.

Author Response

We appreciate the reviewer’s valuable comments on our manuscript and suggestion of publishing after minor revision.  Here are our replies to the comments:

  1. The authors need to improve the image quality. Some images contain text that is too blurred, such as Fig 5c and 6b-6c.

Reply: Thanks for the suggestion from the reviewer. We have improved the quality of images Fig 6c and 7b-c, as well 5b (in revised manuscript).

  1. The introduction is not well presented. I recommend the authors to provide a more detailed introduction about different methods for capturing CTC/ETCs or other cells.

Reply: Thanks for the recommendation from the reviewer. We have added more detailed introduction about different CTC/ETC capture methods, including antibody-based, and antibody-independent methods, such as taSSAW, DLD, DEP, Dean flow technologies as suggested by the reviewers. Proper references directly related to CTC/ETC capture from and outside of reviewer’s suggestions have been added. (see highlighted Line 38-50)

  1. Could this chip be reused again?

Reply: We expect the chip to be one time use because it would be in contact with biohazardous materials after processing patient sample. Parylene deposition, photolithography and reactive ion etching are all scalable manufacturing techniques. We expect the cost of fabricating the parylene membrane can be low if mass produced.

  1. While the manuscript notes improvements over existing techniques, a more concrete comparison with similar designs would provide clearer context and justification for the advancement of this method.

Reply: Thanks for the reviewer’s suggestion. There are other techniques that can fabricate similar designed high porosity parylene membrane, such as micromolding or dry etching using a metal mask, however, more expensive process or equipment is involved. We touched this in the introduction, but now added one more sentence at the beginning of the Section 4 Discussion to provide clearer comparison and justification. Fabrication of high porosity parylene membrane using dry etching with simple photoresist mask has not been reported and is the focus of this manuscript.